# Dynamic Simulation Model-Driven Fault Diagnosis Method for Bearing under Missing Fault-Type Samples

Junqing Ma [1], Xingxing Jiang [2,*], Baokun Han [1,*], Jinrui Wang [1], Zongzhen Zhang [1] and Huaiqian Bao [1]

1    College of Mechanical and Electronic Engineering, Shandong University of Science and Technology, Qingdao 266590, China
2    School of Rail Transportation, Soochow University, Suzhou 215006, China
*    Correspondence: jiangxx@suda.edu.cn (X.J.); bk_han@163.com (B.H.)

**Abstract:** Existing generative adversarial networks (GAN) have potential in data augmentation and in the intelligent fault diagnosis of bearings. However, most relevant studies only focus on the fault diagnosis of rotating machines with sufficient fault-type samples, and some rare fault-type samples may be missing in training in practical engineering. To address those deficiencies, this paper presents an intelligent fault diagnosis method based on the dynamic simulation model and Wasserstein generative adversarial network with gradient normalization (WGAN-GN). The dynamic simulation model of bearing faults is constructed to obtaining simulation signals to replace and complement the missing fault samples, which are combined with the measured signals as training data and then input into the proposed WGAN-GN model for expanding and enhancing the data. To test the effectiveness of the simulated samples, a fault classification model constructed by stacked autoencoders (SAE) is used to classify the enhanced dataset. According to the results, the proposed model performs well when used to diagnose faults under missing samples and is preferable to other methods.

**Keywords:** fault diagnosis; missing samples; dynamic simulation; generative adversarial networks; gradient normalization

## 1. Introduction

In modern industrial production, one of the most important parts of rotating machinery is bearing. Its health status is critical to maintaining the stable operation and safe use of the rotating machinery [1], therefore it is important to develop advance condition monitoring and accurate fault identification of bearings [2]. Due to advances in computer technology, data-driven algorithms based on an increasing number of computer vision and artificial intelligence fields, such as transfer learning [3], support matrix machine [4], graph convolution [5] and convolutional autoencoder [6] have greatly enriched the fault diagnosis methods of rotating machinery. Deep learning-based approaches have strong feature learning capability and can imitate the learning process of the brain by creating deep networks to depict the rich internal information of the data and ultimately achieve accurate fault diagnosis [7–10].

Goodfellow et al. [11] firstly proposed the generative adversarial network (GAN) that can generate new samples in an unsupervised learning way to extract the distribution properties of data. GAN is also widely used in the field of fault diagnosis. Han et al. [12] developed a novel framework for imbalanced fault classification based on Wasserstein generative adversarial networks with gradient penalties (WGAN-GP). Li et al. [13] developed a new ACGAN framework by adding an independent classifier. They introduced the Wasserstein distance and spectral normalization (SN) in the loss functions of GAN. Shao et al. [14] introduced the attention module to guide WGAN-GP to enhance the learning ability. It can be seen that GAN-like models have significant advantages in fault identification and diagnosis due to their characteristics.

It is observed that the above studies are aimed at generating more trustworthy sample data and enhancing training datasets by optimizing and improving the structure of the GAN when all fault-type samples are available for training. However, several types of fault signal data may occasionally be missing, leading to a failure to identify rare fault categories in practical engineering applications. Hence it is essential to build a model to replace and complement the missing fault samples.

To resolve the above deficiencies, a new model-based GAN and dynamic simulation model of fault bearing is proposed in this paper for rotating machinery under missing fault samples. Firstly, to obtain vibration responses of bearings in various states and get the missing fault samples, a rotor-bearing system simulation model is developed. Secondly, gradient normalization (GN) [15,16] is adopted to enhance the feature learning ability of the Wasserstein generative adversarial network (WGAN). The proposed WGAN-GN is used to generate samples for the fault types, and then the generated samples are combined with the original samples into a complete dataset. Next, a stacked autoencoders (SAE) model is employed to extract advanced features from the complete dataset and achieve accurate fault classification. Finally, the viability and robustness of the proposed method are confirmed by several experimental instances. The following is a summary of the main contributions:

(1) A rotor-bearing system simulation model is built to obtain simulation signal of the missing fault type samples.
(2) A novel WGAN-GN method is proposed to generate replaced data under missing sample conditions.
(3) The generated simulated data is joined with the raw data to create a complete dataset for SAE network training to achieve the extraction of features and fault classification.

This paper is organized as follows. The theoretical foundation is explained in Section 2. The framework and steps involved in the proposed method are described in Section 3 in detail. To demonstrate the efficacy of the method, two case studies are applied in Section 4. Finally, the findings are condensed in Section 5.

## 2. Theoretical Background

### 2.1. Construction and Acquisition of Simulation Signal

Some types of fault samples may be missing during the fault diagnosis of mechanical equipment. To simulate this type of situation and obtain the lacked fault samples, the concentrate quality standard was established with the rolling bearing fault of the rotor-bearing-casing coupling system dynamics model. This was followed by the establishment of the dynamic model of the rolling bearing fault and the construction of the lack of fault samples. As shown in Figure 1, a simulation model made up of a rotor, bearing, bearing seat, shaft, and other parts are depicted. The bearing inner ring remains connected to the bearing housing while the shaft moves. Suppose the stiffness and damping of the primary components are not changed numbers and the left bearing occurs faults, six differential equations of motion pertaining to the left support bearing can be listed [17]:

$$m_{bL}\ddot{x}_{bL} + k_{fLH}(x_{bL} - x_c) + c_{fLH}(\dot{x}_{bL} - \dot{x}_c) + k_{tLH}(x_{bL} - x_{wL}) + c_{tLH}(\dot{x}_{bL} - \dot{x}_{wL}) = 0 \tag{1}$$

$$m_{bL}\ddot{y}_{bL} + k_{fLV}(y_{bL} - y_c) + c_{fLV}(\dot{y}_{bL} - \dot{y}_c) + k_{tLV}(y_{bL} - y_{wL}) + c_{tLV}(\dot{y}_{bL} - \dot{y}_{wL}) = -m_{bL}g \tag{2}$$

$$m_{rL}\ddot{x}_{rL} + k(x_{rL} - x_{rp}) + c_{rb}\dot{x}_{rL} - F_{xbL} = 0 \tag{3}$$

$$m_{rL}\ddot{y}_{rL} + k(y_{rL} - y_{rp}) + c_{rb}\dot{y}_{rL} - F_{ybL} = -m_{rL}g \tag{4}$$

$$m_{wL}\ddot{x}_{wL} + k_{tLH}(x_{wL} - x_{bL}) + c_{tLH}(\dot{x}_{wL} - \dot{x}_{bL}) + F_{xbL} = 0 \tag{5}$$

$$m_{wL}\ddot{y}_{wL} + k_{tLH}(y_{wL} - y_{bL}) + c_{tLH}(\dot{y}_{wL} - \dot{y}_{bL}) + F_{ybL} = -m_{wL}g \tag{6}$$

where, $m_{bL}$ is the mass of bearing support; $k_{fLH}$ and $k_{fLV}$ are the transverse and longitudinal support stiffness between the shell and the bearing support respectively. $c_{fLH}$ and $c_{fLV}$ are the transverse and longitudinal support damping between the shell and the bearing

support respectively. $k_{tLH}$ and $k_{tLV}$ are the transverse and longitudinal support stiffness between the bearing outer ring and the bearing support respectively. $c_{tLH}$ and $c_{tLV}$ are the transverse and longitudinal extrusion film damping between the bearing outer ring and the bearing support. $m_{rL}$ is the equivalent mass of rotor. $k$ and $c_{rb}$ are the shaft stiffness and damping of the rotor at the bearing. $F_{xbL}$ and $F_{ybL}$ are the supporting reaction forces of the bearing, and $m_{wL}$ is the mass of the outer ring of the bearing.

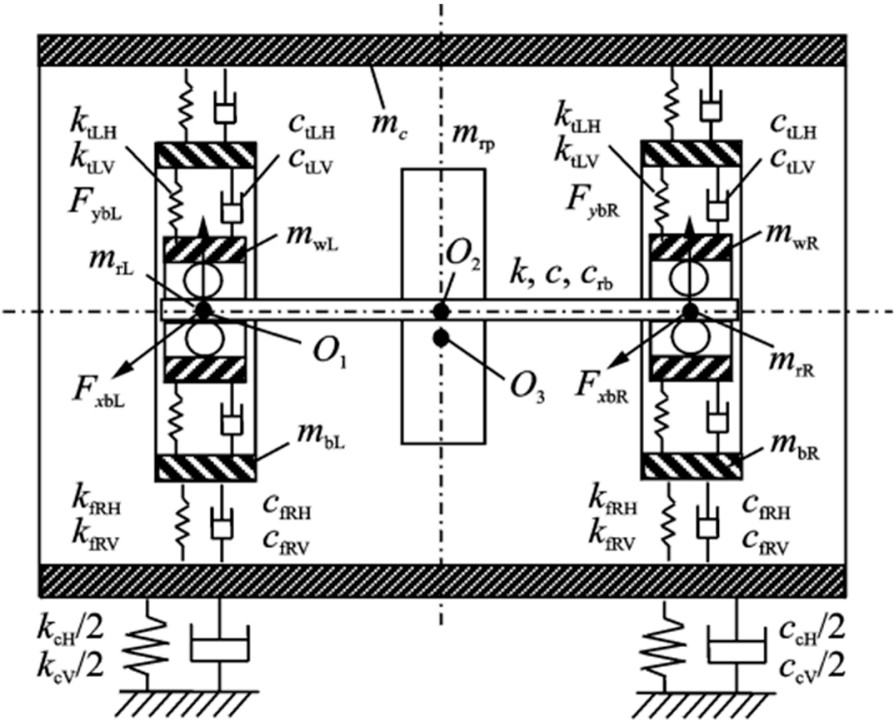

**Figure 1.** Diagrammatically representing the simulated rotor-bearing system.

A schematic illustration of a bearing with a localized defect area is shown in Figure 2 on the ball, inner ring, and outer ring components. The cyclically varying contact forces $F_{xbL}$ and $F_{vbL}$ are the sum of the contact forces of the balls at different angles:

$$F_{xbL} = \sum_{j=1}^{Z} f_j cos\theta_j, F_{ybL} = \sum_{j=1}^{Z} f_j sin\theta_j \qquad (7)$$

where $\theta_j$ is the angular position of the jth ball, and $f_j$ is the contact force between the jth ball and the raceway calculated according to the following formula:

$$f_j = C_b [\delta_j]^n = C_b (xcos\theta_j + ysin\theta_j - \delta_0 - \delta_D)^n \times H(xcos\theta_j + ysin\theta_j - \delta_0 - \delta_D) \quad (8)$$

where $C_b$ is the Hertzian contact stiffness, $H(\cdot)$ is the Heaviside function, $\delta_j$ is the clearance between the ball and the raceway, $\delta_0$ is the initial clearance, $\delta_D$ is the clearance caused by the local defect area, and $n$ is the load-deformation coefficient, which is 2/3 for the ball bearing.

The simulation model adopts a SKF6203 bearing and its main parameters are listed in Table 1. The damage diameter $L_D$ is set to 0.3556 mm, the damage depth a is set to 2.794 mm, the rotational speed is 1797r/min, and the sampling frequency is set to 10 kHz. Figure 3 shows a time-domain waveform diagram of the vibration acceleration signal collected by the left support bearing under four different health states.

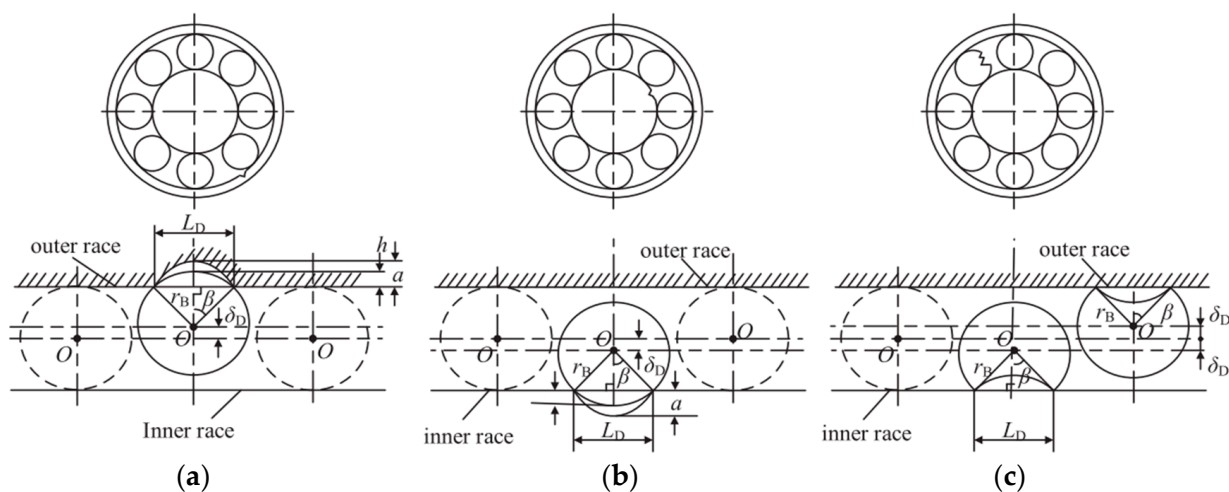

**Figure 2.** Schematic of the local defect area: (**a**) Outer race fault, (**b**) Inner race fault, (**c**) Ball fault.

**Table 1.** Main parameters of 6203.

| Description of Parameters | Values of Parameters |
| --- | --- |
| The radius of outer race/mm | 17.0 |
| The radius of inner race/mm | 39.9 |
| Pitch diameter/mm | 28.3 |
| Diameter of rolling element/mm | 6.8 |
| Number of balls | 8 |
| Contact angle/° | 0° |

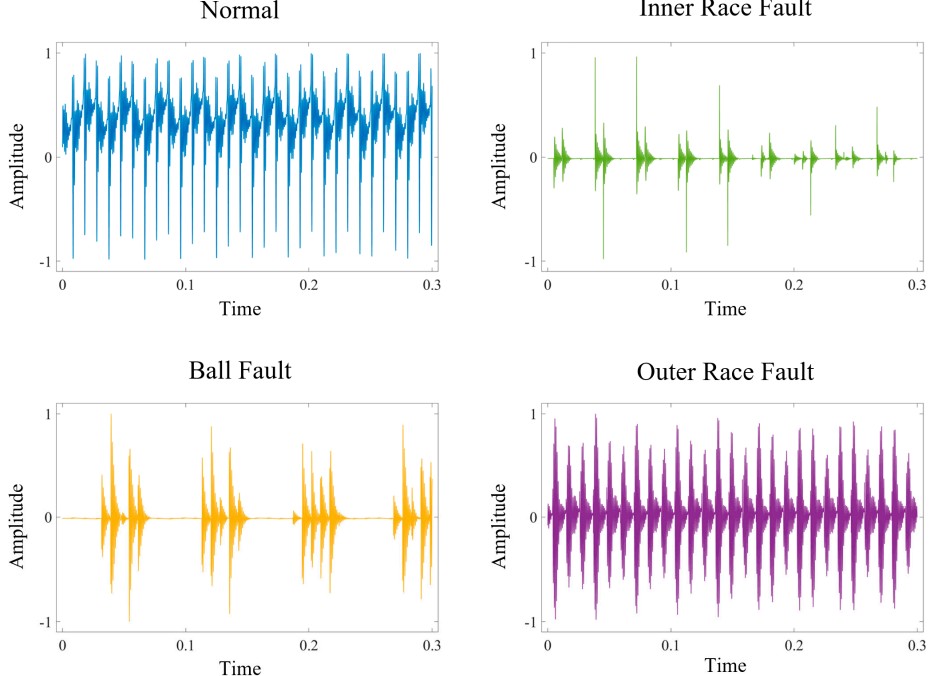

**Figure 3.** Time−domain waveforms of simulation data.

## 2.2. Wasserstein Generative Adversarial Networks with Gradient Normalization (WGAN-GN)

The Generator (G) and Discriminator (D) are the two components that make up the GAN structure, as shown in Figure 4. A random noise vector is utilized as the input of the G, which attempts to create realistic data to deceive D. The D then learns to distinguish between actual data and G-produced synthetic data. Typically, G and D are constantly

optimized to enhance their ability to generate and discriminate data, and they are often parameterized as deep neural networks and optimize a mini-max objective. The training objective function of the GAN is displayed as follows:

$$\min_{G}\max_{D}\mathbb{E}_{x\sim p_r(x)}[\log(D(x))] + \mathbb{E}_{\tilde{x}\sim p_g(x)}\left[log\left(1 - D\left(\tilde{x}\right)\right)\right] \tag{9}$$

where $p_r(x)$ is the distribution of raw data and $p_g(x)$ is the distribution defined by $p_g = G_*(p_z)$, * is the push-forward measure and $p_z$ is the distribution of stochastic noise vector.

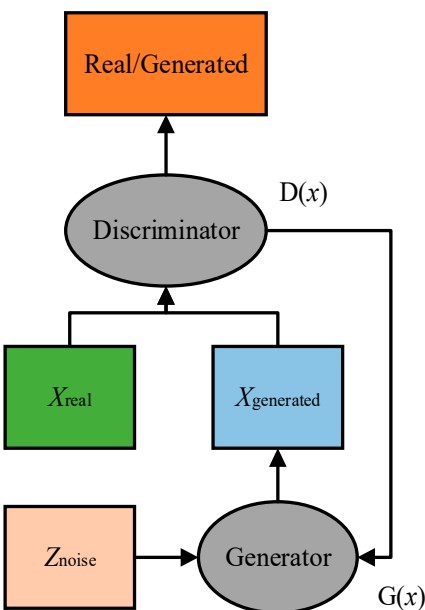

**Figure 4.** Generative adversarial networks.

If D is always optimal in this scenario, the G will inevitably converge to the true distribution $p_r(x)$. However, gradient vanishing and gradient explosion are the two fundamental problems that training GAN always faces. Optimizing Equation (1) is comparable to minimizing the Jensen-Shannon divergence (JS-divergence) between $p_g(x)$ and $p_r(x)$ for these unstable training situations. If $p_g(x)$ and $p_r(x)$ do not overlap, the gradient would vanish since the JS-divergence would be a constant number. Second, D frequently succumbs to overfitting with finite real samples, which subsequently results in gradient explosion near real samples. So the JS-divergence may be not a suitable cost function.

A series of recent studies have focused on solving the problem of unstable training. The steep gradient space of the discriminator is one of the causes of unstable GAN training, which can cause a pattern crash during the training of the generator. Although simple methods such as L2 normalization and weight clipping can effectively make the GAN training process more stable, the discriminator model capacity is constrained by these extra restrictions. So instead of learning to produce true data, the generator is more likely to trick the discriminator. The regularization or normalizing of the discriminator is another well-liked method for formalizing the discriminator as a Lipschitz continuous function. This way allows for the smoothing of the discriminator gradient space without compromising discriminator speed.

The purpose of WGAN is to minimizing the Wasserstein distance between $p_g(x)$ and $p_r(x)$, i.e.,

$$\min_{G}\max_{D, L_D \leq 1}\mathbb{E}_{x\sim p_r(x)}[D(x)] - \mathbb{E}_{\tilde{x}\sim p_g(x)}\left[D\left(\tilde{x}\right)\right] \tag{10}$$

where $L_D$ is the Lipschitz constant of discriminator D. $L_D$ is defined as follows:

$$L_D inf := L \in \mathbb{R} : |D(x) - D(y) \leq L \parallel x - y \parallel, \forall x, y \in \mathbb{R}^n \tag{11}$$

where $\| \cdot \|$ can be the norm of vector.

By maximizing Equation (2) under the Lipschitz constraint in Equation (3), the discriminator in WGAN seeks to approach the Wasserstein distance. The generator performs better at simulating the real distribution if the discriminator can operate in a bigger function space because it can estimate the Wasserstein distance with greater accuracy. The Lipschitz constraint, meanwhile, reduces the steepness of the value surface and lessens the overfitting of the discriminator. In comparison to KL-divergence and JS-divergence, Wasserstein distance provides superior smoothing properties. The vanishing gradient issue can be theoretically resolved.

Three characteristics can be used to define the imposition of a Lipschitz constraint on the discriminator.

(1) Constraint on a model or module. Model-level restriction, in our opinion, is preferable to module-level constraint because it will limit the model capacity of layers, drastically lowering the potential of neural networks.
(2) Constraint that are sample-based or not. The non-sampling-based method performs better than the sampling-based method since the latter may not be applicable to data that has not already been sampled.
(3) Firm or flexible restriction. Since the continuous Lipschitz constant ensures gradient stability against unobserved data, the hard constraint outperforms the soft constraint.

How to achieve the Lipschitz constraint is a problem that needs to be focused on in neural networks, because it is difficult to maintain a great balance between the Lipschitz constraint and network capacity. Many methods have been proposed to achieve this constraint, but most of them cannot meet the above three conditions. It has been demonstrated that the Lipschitz constant of layer level Lipschitz constraints (such as SN-GAN) may drastically drop when the number of layers increase. The ideas of parameter clipping and spectral normalization (SN) are similar. They both guarantee that the Lipschitz constant of each layer is bounded by constraining parameters, so the total L constant is also bounded. The gradient penalty ensures that the "soft constraint" is imposed through the penalty term. Therefore, WGAN-GN is proposed as shown in Figure 5. To be specific, ReLU or LeakyReLU is usually adopted as the activation function. It is a "piecewise linear function" under this activation function, which means that it is a linear function in the local continuous region, except for the boundary. Correspondingly, it is also a constant vector. The loss function is as follows:

$$\hat{D} = D(x)/(\| \nabla D(x) \| + |D(x)|) \tag{12}$$

$$L(D) = \mathbb{E}_{\tilde{x} \sim p_r(x)}\left[\hat{D}(D(x))\right] - \mathbb{E}_{\tilde{x} \sim p_g(x)}\left[\hat{D}(D(\tilde{x}))\right] \tag{13}$$

$$L(G) = -\mathbb{E}_{x \sim p_z(x)}\left[\hat{D}(G(x))\right] \tag{14}$$

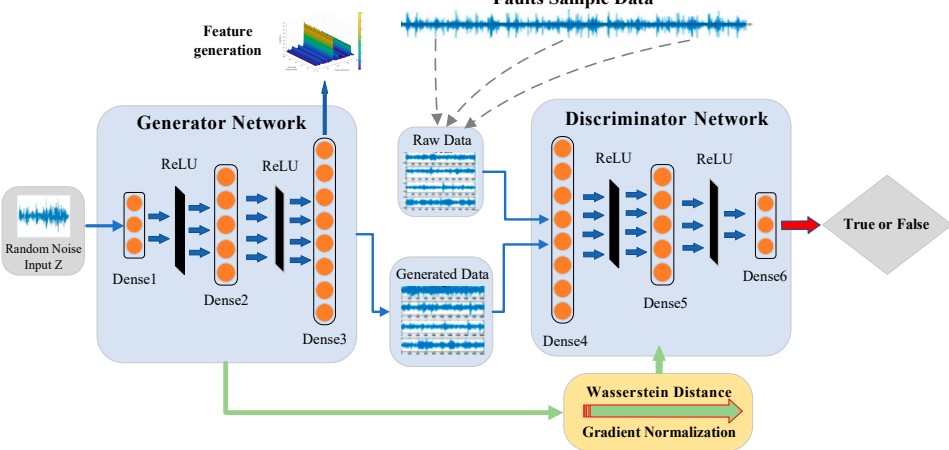

**Figure 5.** Structure of WGAN-GN.

Among them, $\hat{D}$ is the custom function, $D(x)$ is the independent variable ($G(x)$ or $D(x)$), $\| \nabla D(x) \|$ is the any norm of a vector.

The employed gradient normalization (GN) module has few hyperparameters and is easy to apply to different rotating machines without substantial modifications.

## 3. System Framework and Model Training

### 3.1. System Framework Design

The generator in WGAN-GN is made up of four completely connected networks with 10, 50, 100, and 200 neurons per layer, respectively. The discriminator has four fully connected layers as well with 100, 50, 10, and 1 neurons per layer. ReLU is chosen as the activation function, with the dimension of random noise set to 10. The SAE module is used to classify faults, and the structure contains three layers with 200, 100, and 4 neurons. The SAE-connected classifier is then implemented using Softmax regression. The SAE weights are updated by the BP algorithm, which also allows parameters to be adjusted. The adaptive moment estimation (ADAM) optimization algorithm is used for model updating.

### 3.2. Model Training Procedure WGAN-GN-SAE

(1)  The system dynamics model of rolling bearing is established to perform the bearing fault modeling, and the missing fault simulation vibration signal of rolling bearing is obtained.
(2)  The signal is pre-processed by fast Fourier transform (FFT) and Hilbert transform to acquire the envelope signal, then the training and testing data are equally separated.
(3)  The training data is input into WGAN-GN for data enhancement.
(4)  The simulated data generated by WGAN-GN are coupled with the original data to enhance the dataset and form a complete fault dataset.
(5)  The complete fault dataset is used as training data of the SAE network, and the testing data are used for model testing.

## 4. Experimental Verification

### 4.1. Case 1: Bearing Dataset with One Missing Failure Sample

The rolling bearing dataset used in this section is from Case Western Reserve University (CWRU). As shown in Figure 6, the test bench includes a 2.72 kW motor, a torque sensor, a power test meter, and an electronic controller (as shown in the Figure 7) [18]. The bearings to be tested support the motor shaft, the drive side bearing is SKF 6205 and the fan side bearing is SKF6203. The 6203 deep groove ball bearing is chosen for the roller bearing study in this work, and the dataset includes normal condition (NC), outer ring fault (OF), inner ring fault (IF), and roller element fault (RF). The fault is handled by electrical discharge machining (EDM) single point damage.

#### 4.1.1. Data Pre-Processing

In this section of experiments, the signals of bearing outer ring fault (6:00 position) obtained from the dynamic simulation model, inner ring fault and rolling element fault from real measurement are investigated. Firstly, in order to verify that a good diagnosis can be achieved in the proposed method in this paper, the dataset in Tables 2 and 3 is used as the experimental data. 200 samples are randomly selected from each health state, then the envelope signal of the sample is obtained, each sample contains 200 envelope signal data points. Half of the samples are used as training data and the other half as test data. In dataset B, the training data are the signal obtained from the simulation model and the test data are the signal obtained from the real measurement in the outer ring fault type.

#### 4.1.2. Generate Visual Evaluation of the Sample

The Envelope signals of the 4 fault states were generated by applying the proposed WGAN-GN model. Figure 8 gives the comparison of the generated and real samples. It is clear that the feature trends are very similar. There are some differences in the amplitude,

indicating that the WGAN-GN has learned the distribution of the real data, and the generated samples are sampled from the real distribution instead of simply copying the original samples, which can enhance the robustness and generality of the proposed method. To put it another way, WGAN-GN has taken the major features from the original signal and removed some noisy components that are bad for pattern recognition.

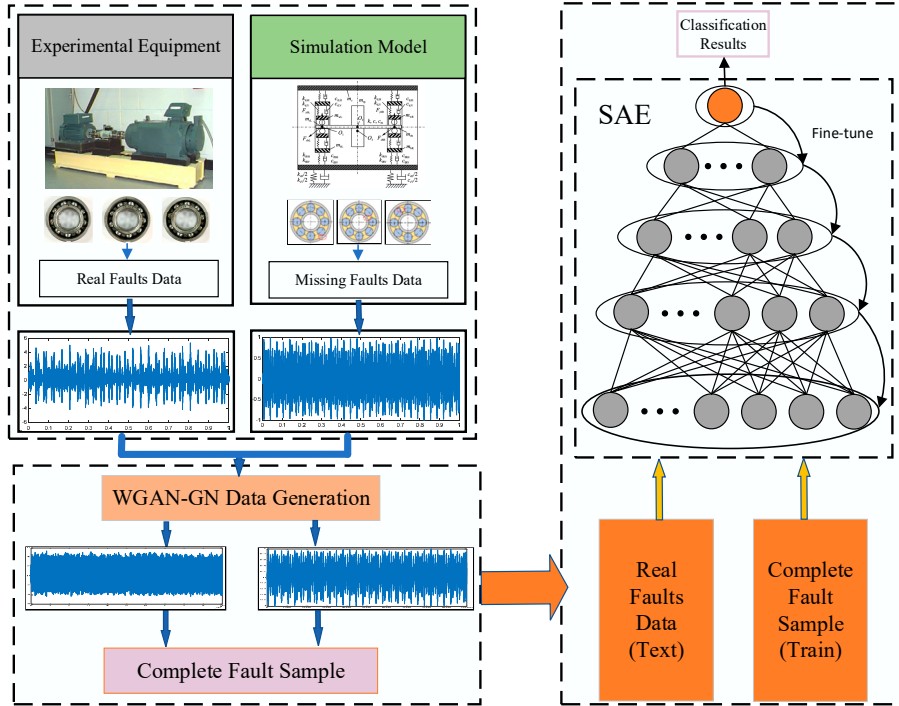

**Figure 6.** Fault diagnosis model design.

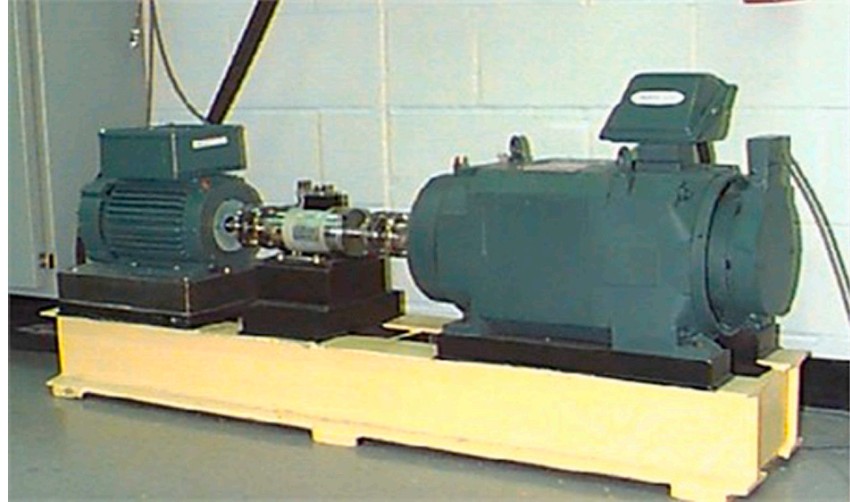

**Figure 7.** Rotating machinery fault diagnosis experimental device.

**Table 2.** Situation of Dataset A.

| Category | Fault Location | Signal Source Dataset B | Sample Size (Train/Test) |
|---|---|---|---|
| NC | Normal | Measurement | 100/100 |
| RF | Ball | Measurement | 100/100 |
| IF | Inner Race | Measurement | 100/100 |
| OF | Outer Race | Measurement | 100/100 |

**Table 3.** Situation of Dataset B.

| Category | Fault Location | Signal Source Dataset B | Sample Size (Train/Test) |
|---|---|---|---|
| NC | Normal | Measurement | 100/100 |
| RF | Ball | Measurement | 100/100 |
| IF | Inner Race | Measurement | 100/100 |
| OF | Outer Race | Simulation | 100/100 |

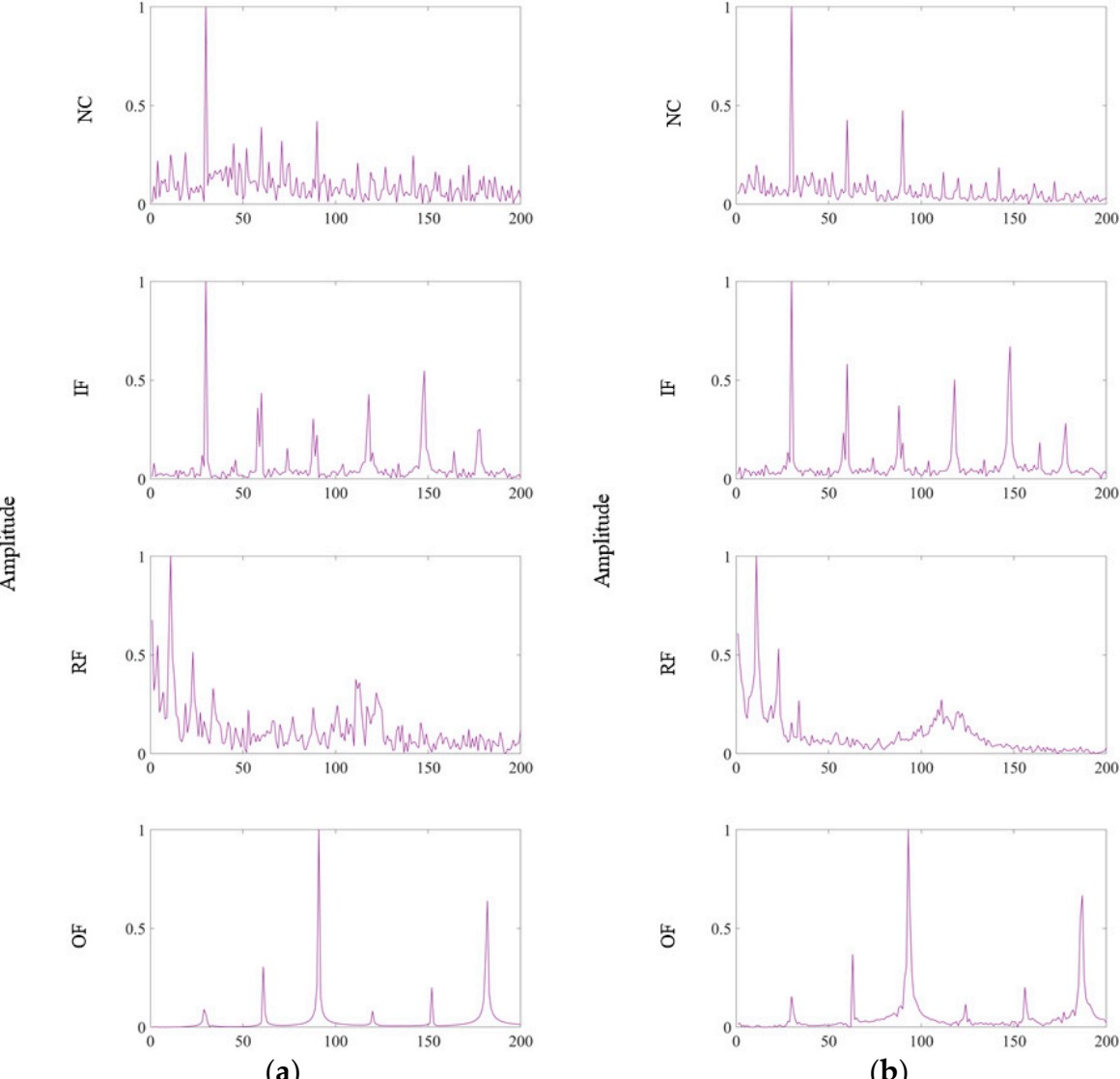

**Figure 8.** Comparison between raw and generated signals for bearings: (**a**) Raw signals, (**b**) Generated signals.

It becomes vital to look into the feature learning process in order to further demonstrate the feature extraction capacity of WGAN-GN. The extraction process for all the NC tomographic samples is displayed in the 3D space as shown in Figure 9. It is seen that the variation amplitude becomes larger and the characteristics become more and more obvious as the network layer deepens.

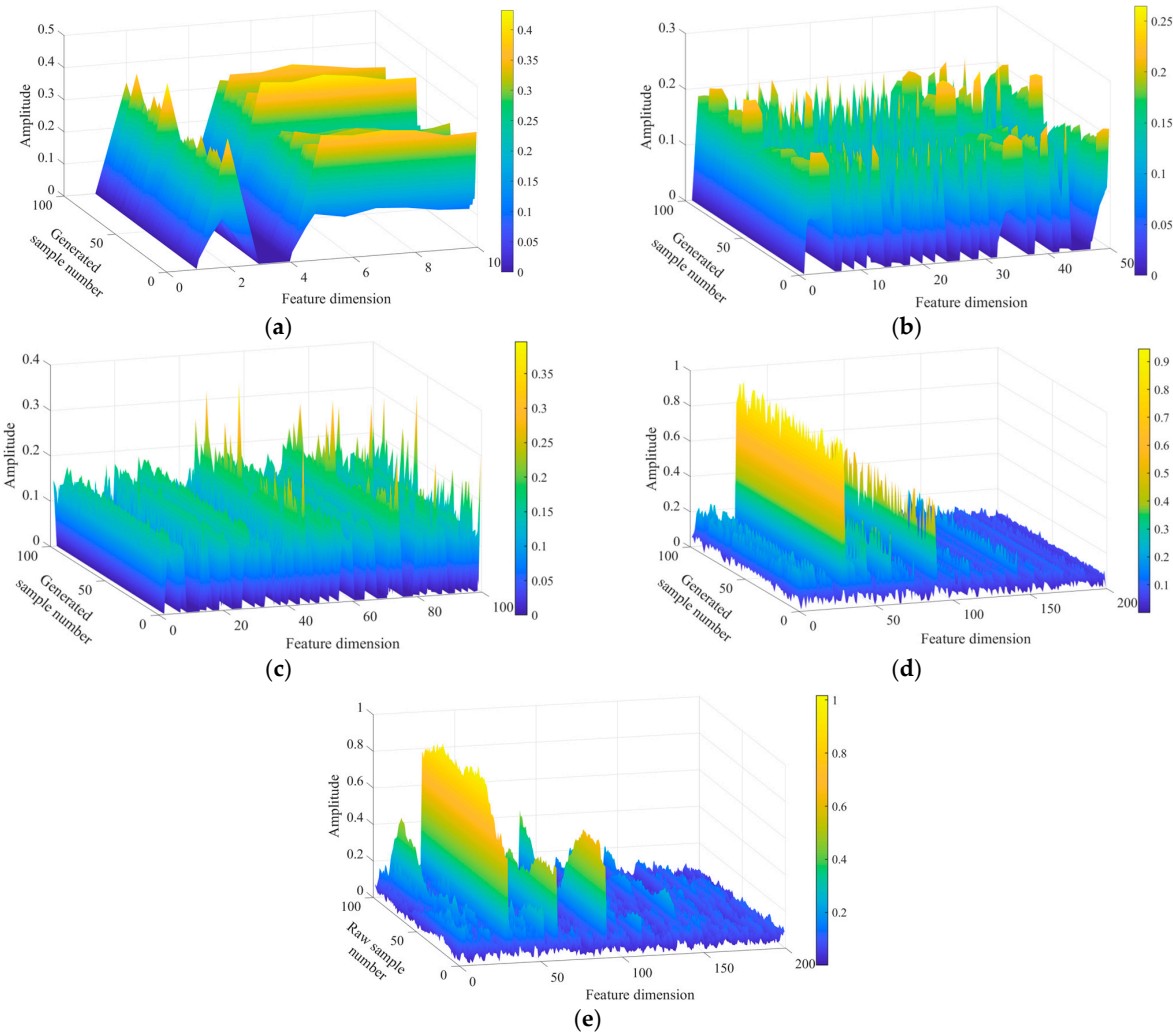

**Figure 9.** Extracted features of the generator for NC samples: (**a**) the 1st layer, (**b**) the 2nd layer, (**c**) the 3rd layer, (**d**) the output layer, (**e**) the raw sample.

### 4.1.3. Comparison of Neural Network Model Training Results

To accurately demonstrate how superior the proposed approach is. Each method received twenty experiments, and the classification accuracy results of Datasets A and B are displayed in Figure 10. For Dataset A, the average classification accuracy and standard deviation of WGAN-GN are 97.45% and 0.27%, respectively, and the two corresponding values are 91.27% ± 1.01% for WGAN and 81.19% ± 1.49% for GAN. The average accuracy of GAN is 81.78% ± 1.16% for Dataset B, while the average accuracy of WGAN is 90.88% ± 1.05%. In contrast, the average accuracy of WGAN-GN is 96.95% ± 0.49%. The comparison shows that the proposed WGAN-GN is significantly better than other methods. The strategy proposed in this study can improve the accuracy of diagnosis to a certain extent when the real measured signals are used as the data set. However, when the simulation signal obtained from the kinetic model is used as one of the fault signals, the experimental accuracy is also not significantly decreased, indicating that the obtained simulation signal can replace the real measured signal to a certain extent for fault diagnosis.

T-SNE [19] is used for the visual operation of dimension reduction, and the results of Dataset A are shown in Figure 11. As be observed, GAN is only able to appropriately identify RF and OF samples, and there are different degrees of confusion among other fault samples. As Figure 10b, WGAN has some classification ambiguity, but the aggregation among samples is relatively good. In contrast, the proposed WGAN-GN fully isolates all bearing failure samples, and the samples of the same type are strongly aggregated, which

illustrates the efficiency of the proposed method. In addition, when using simulation signal as the training data as in Figure 12, WGAN-GN still performs far better than the other two approaches. As a result, it is concluded that, the constructed model has the best ability for fault classification compared to the other methods. Meanwhile it also proves that the obtained simulation signal can replace the real measured signal to a certain extent for fault diagnosis.

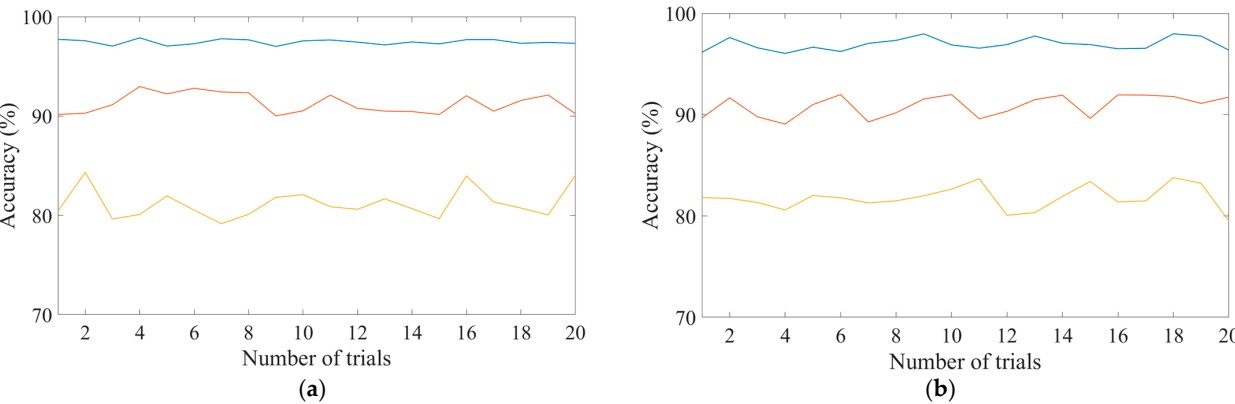

**Figure 10.** Accuracies of the two datasets using the three methods: (**a**) Dataset A, (**b**) Dataset B.

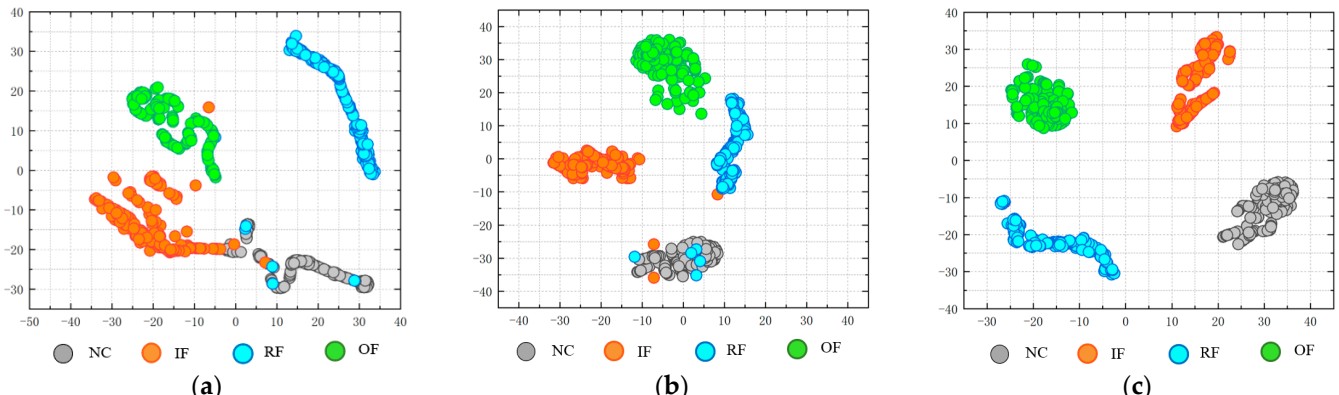

**Figure 11.** Dimensionality reduction visualization results on Dataset A: (**a**) GAN, (**b**) WGAN. (**c**) WGAN-GN.

### 4.2. Case 2: Bearing Dataset with Two Missing Failure Samples

The dataset used in this section is still from Case Western Reserve University, the difference is that in order to further verify the efficiency of the proposed method, increasing the number of emulated signals. 200 samples are also randomly selected from each health state, then the envelope signal of the sample is obtained. Half of the samples are used as training data and the other half as test data. The details of the Dataset C are listed in Table 4. This section selects the inner ring fault and outer ring fault as the missing fault samples, and the missing fault samples with simulated signals are replaced for the experiments.

**Table 4.** Situation of Dataset C.

| Category | Fault Location | Signal Source Dataset C | Sample Size (Train/Test) |
|---|---|---|---|
| NC | Normal | Measurement | 100/100 |
| RF | Ball | Measurement | 100/100 |
| IF | Inner Race | Simulation | 100/100 |
| OF | Outer Race | Simulation | 100/100 |

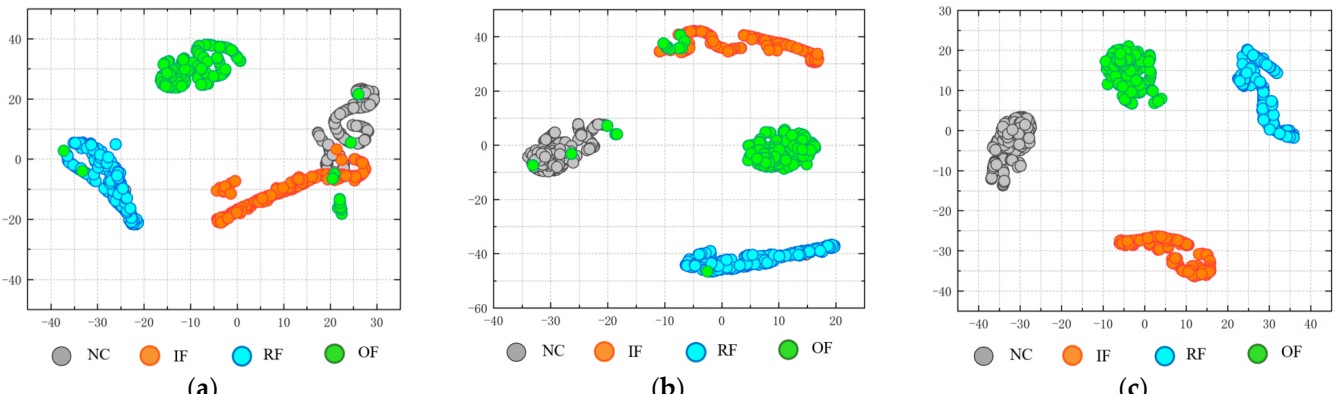

**Figure 12.** Dimensionality reduction visualization results on dataset B: (**a**) GAN. (**b**) WGAN. (**c**) WGAN-GN.

Diagnosis Results

Figure 13 gives the comparison result of Dataset C. It is obvious that the trends of the generated samples and the raw samples are very similar, and the features in the generated samples have basically the same distribution as the features in the raw samples. By contrasting the generated samples with the original samples of 3D plots, RF samples are chosen to investigate the feature extraction capability of WGAN-GN. According to Figure 14, the simulated signals generated by WGAN-GN has nearly identical properties to the original samples.

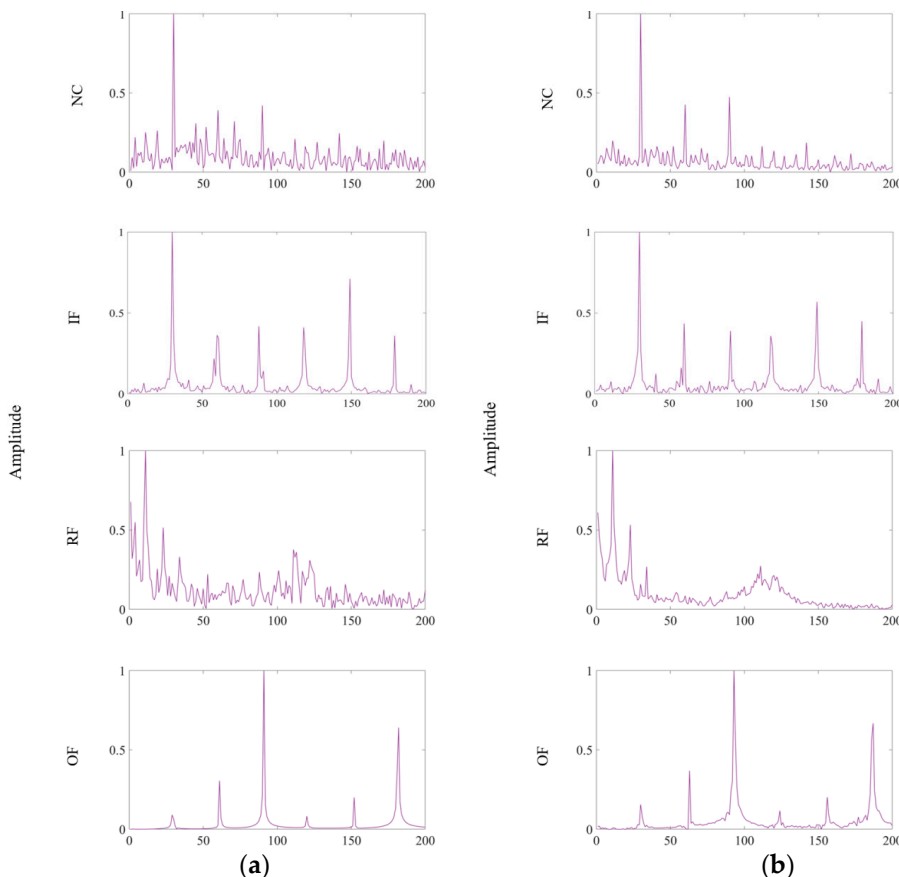

**Figure 13.** Comparison between real and generated signals for dataset C: (**a**) Raw signals, (**b**) Generated signals.

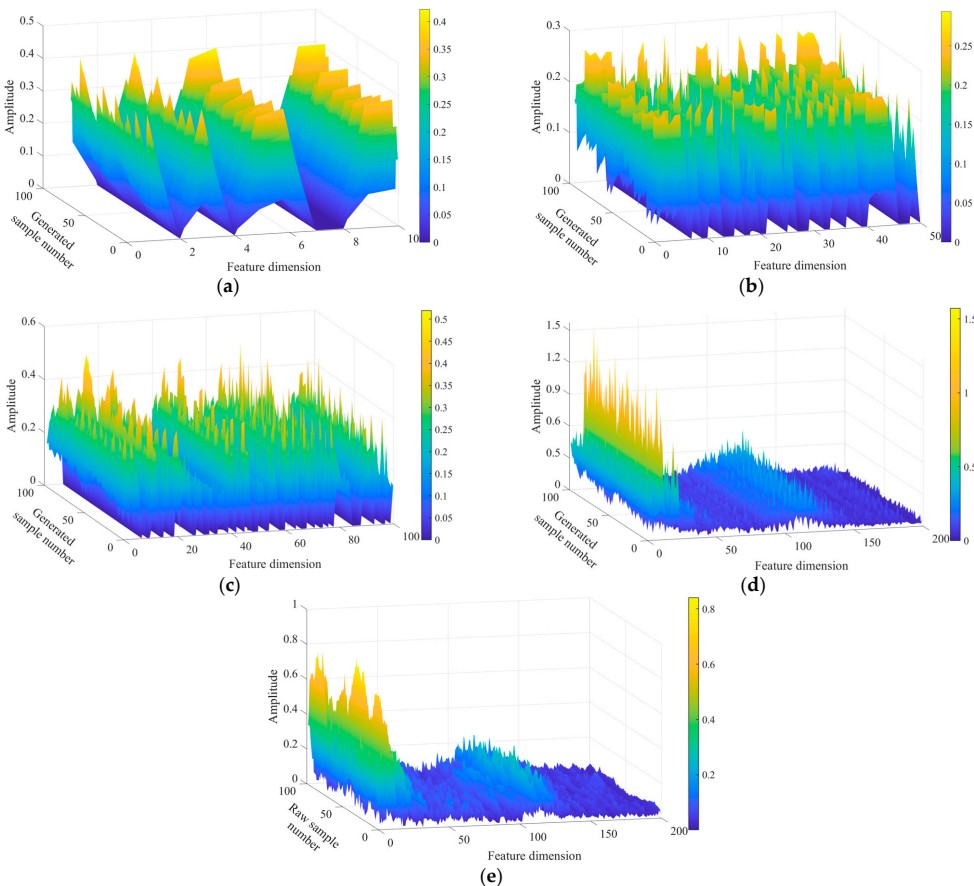

**Figure 14.** Extracted features of generator for RF samples: (**a**) The first layer, (**b**) The second layer, (**c**) The third layer, (**d**) The output layer, (**e**) The raw sample.

Figure 15 displays the findings of the diagnosis accuracy test. For Dataset C, the average accuracies of the three methods are 81.37%, 90.00%, and 96.48%, respectively, with 1.61%, 1.01%, and 0.85% as their standard deviations. The proposed approach continues to offer maximum degree of diagnostic accuracy when compared to other methods. Furthermore, Figure 16 displays the outcomes of the dimension reduction. It is concluded that the proposed model greatly outperforms them in terms of clustering and classification compared to the other two methods. It also proves that the obtained simulation signal can replace the real measured signal to a certain extent for fault diagnosis at the same time.

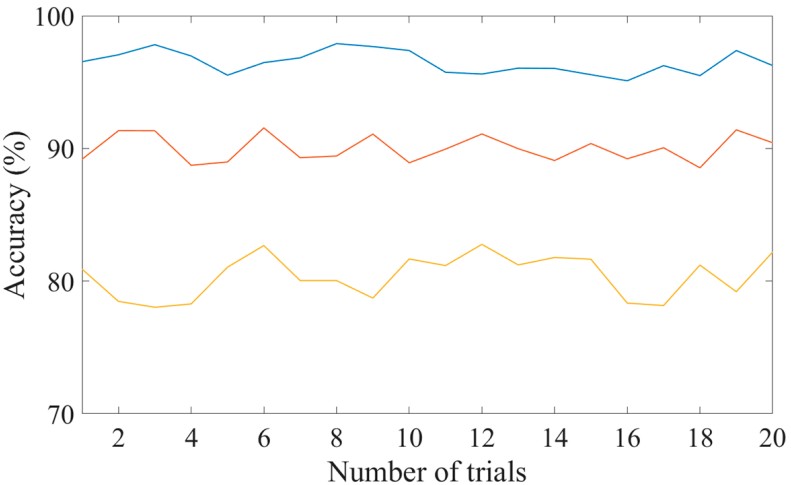

**Figure 15.** Accuracies of the datasets C using the three methods.

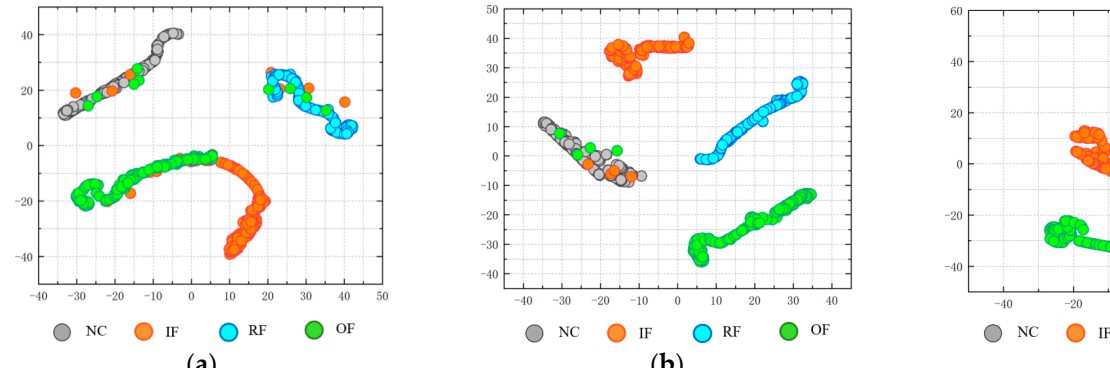

**Figure 16.** Dimensionality reduction visualization results on Dataset C: (**a**) GAN. (**b**) WGAN. (**c**) WGAN-GN.

## 5. Conclusions

In this study, a new method based on dynamic simulation model and WGAN-GN is proposed as a solution for fault diagnosis of missing experimental fault samples of bearing. In the proposed method, a dynamic simulation model is first built to obtain simulated data of missing fault type, which can be used to form relatively complete fault samples with other signals. Then WGAN-GN is employed to generate expanded samples from a smaller number of relatively complete fault samples. Next, these simulated samples are combined with the original samples to form an enhanced fault dataset which can be called as complete fault dataset. Finally, SAE is used to classify the complete fault samples. Three datasets of rotating machines were used to confirm the validity of the method. The following is a summary of the main conclusions:

1.  It is demonstrated that the developed dynamic simulation model can generate high-quality replacement samples with missing fault samples to some extent.
2.  The effective feature extraction and data generation capability of the proposed model is illustrated by the features learned continuously from the hidden layer of WGAN-GN.
3.  The experimental results show that using the proposed method can help to improve the accuracy of diagnosis when the types of fault sample data are insufficient.
4.  Both the applicability to other mechanisms and the problem regarding the in-fluence of noise are part of our future research objectives.

**Author Contributions:** Conceptualization, J.M.; methodology, X.J.; software; J.W.: writing—review and editing, B.H.; funding acquisition, X.J.; supervision, Z.Z.; formal analysis, H.B. All authors have read and agreed to the published version of the manuscript.

**Funding:** This research was funded by the [China Postdoctoral Science Foundation] grant number [2021M70275, 2022T150552], [Prospective Application Research of Suzhou] grant number [SYG202111], [National Natural Science Foundation of China] grant number [52005303, 52105110], [Natural Science Foundation of Shandong Province] grant number [ZR2022ME119, ZR2020QE157, ZR2021QE024].

**Institutional Review Board Statement:** Not applicable.

**Informed Consent Statement:** Not applicable.

**Data Availability Statement:** The experimental data used in this paper is from Western Reserve University, which can be found at the following website: http://www.52phm.cn/blog/detail/54 (accessed on 4 December 2021).

**Acknowledgments:** The authors are very grateful for all constructive comments that helped us improve the original version of the manuscript.

**Conflicts of Interest:** The authors declare no conflict of interest.

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
