# Peer review of "Dynamic Simulation Model-Driven Fault Diagnosis Method for Bearing under Missing Fault-Type Samples"

_applsci, doi:10.3390/app13052857_

Round 1
Reviewer 1 Report
The paper is well organized and well written. However, it lacks a novelty and contribution for these reasons :
1. most of analysis was preseneted previously by other authors
Li, W., Zhong, X., Shao, H., Cai, B., & Yang, X. (2022). Multi-mode data augmentation and fault diagnosis of rotating machinery using modified ACGAN designed with new framework. Advanced Engineering Informatics, 52, 101552.
2. Eqs. (1:6) have not been discussed and solved in the article. So, they should be removed.
3. Figure 6 in this article is completely presented in Figure 4 in the previous article, without any copyright.
4. The details of bearing samples, dimensions, fault not included in the article.
Therefore, I reject the article.
Author Response
Response to Reviewer 1 Comments
Thank you for offering us an opportunity to improve the quality of our submitted manuscript. We appreciated very much your constructive and insightful comments. In the following text, our point-to-point responses to the queries raised are listed.
We highlighted all the revisions and responses in red color.
Point 1: Most of analysis was presented previously by other authors.
Li, W., Zhong, X., Shao, H., Cai, B., & Yang, X. (2022). Multi-mode data augmentation and fault diagnosis of rotating machinery using modified ACGAN designed with new framework. Advanced Engineering Informatics, 52, 101552.
Response 1: Thank you very much for the careful suggestion. Next we will briefly describe the differences between this paper and previous study.
Li et al. developed a new ACGAN framework by adding an independent classifier. They introduced the Wasserstein distance and spectral normalization (SN) in the loss functions of Generative adversarial networks (GAN), which can efficiently generate fault samples. And it is clear that they were aimed at generating more trustworthy sample data and enhancing training datasets by optimizing and improving the structure of the GAN when all the fault type samples are available for training. However, in our paper we wanted to address fault diagnosis for bearings when several types of fault signal data are missing instead of only simply generating and enhancing datasets. Firstly, we established the dynamic model of the rolling bearing and obtained the missing fault types simulation vibration signals of rolling bearing. In addition, we firstly introduced the gradient normalization (GN) and Wasserstein distance into the GAN for fault diagnosis instead of SN, which can also generate sample data with higher qualities. We believe that can provide some new methods and ideas for later researchers.
In other words, both the problem to be solved and the method proposed in this paper are different from the previous one. But the previous article still provides us with valuable experience and reference.
Point 2: Eqs. (1:6) have not been discussed and solved in the article. So, they should be removed.
Response 2: In this paper we wanted to address fault diagnosis for bearings when several types of fault signal data are missing. So to obtain the missing fault types simulation vibration signals of rolling bearing, we need to establish the dynamic model. So using differential equations to build dynamic models is one of the several focuses of this paper. Eqs. (1:6) are the six differential equations of motion representing the rolling bearing dynamics model. In addition, the text and equations in section 2.1 are all explanations and additions to Eqs. (1:6). So we think this section should be kept.
Point 3: Figure 6 in this article is completely presented in Figure 4 in the previous article, without any copyright.
Response 3: This dataset is a public dataset and has been added to the text by reference. (Line 252, Page 8)
[18]Smith, W. A.; Randall, R. B. Rolling element bearing diagnostics using the Case Western Reserve University data: A benchmark study. Mech Syst Signal Pr. 2015: 64, 100-131.
Point 3: The details of bearing samples, dimensions, fault not included in the article.
Response 4:Thank you very much for your valuable and professional suggestion.The details of bearing have benn included as follows in red.
The simulation model adopts a SKF6203 bearing, and its main parameters are listed in Table 1. The damage diameter is set to 0.3556 mm, the damage depth a is set to 2.794 mm, the rotational speed is 1797r/min, and the sampling frequency is set to 10 kHz. Figure. 3 shows a time-domain waveform diagram of the vibration acceleration signal collected by the left support bearing under four different health states.(Line 118-122,Page 2)
Please refer to the attachment for specific modifications.
We tried our best to improve the manuscript and we appreciate for your warm work earnestly, and hope that the correction will meet with approval.
Thanks again for your comments and suggestions.

Reviewer 2 Report
The article is very well written. It is legible. A very interesting and extensive topic. The topic of the article is very topical. The article has been written in a systematic manner with logical continuity. The obtained results do not raise any objections. Their presentation is correct and complete. There are errors in the text (see below). I do not make substantive comments. I have notes for editing.
1) Line 99
It is: „… respectively. ctLHand ctLV …”
Should be: „… respectively. ctLH and ctLV …”
2) Line 101
It is: „… mass of rotor; ? and ??? are …”
Should be: „… mass of rotor. ? and ??? are …”
3) Line 102
It is: „… the bearing, ???? and ???? are …”
Should be: „… the bearing. ???? and ???? are …”
4) Line 114
It is: „… area, and n is the load-deformation…”
Should be: „… area, and n is the load-deformation…”
5) Line 129, 133, 146, 209, 354
Put a period at the end of the line.
6) Line 132
It is: „… defect area. (a) Outer race fault. (b) Inner race fault. (c) Ball fault.”
Should be: „… defect area: (a) Outer race fault, (b) Inner race fault, (c) Ball fault.”
7) Line 134, 220, 234, 246, 259, 323, 333,
Add a dot after the numbering.
8) Line 180
Start scoring „1)” on a new line.
9) Line 182
Start scoring „2)” on a new line.
10) Line 185
Start scoring „3)” on a new line.
11) Line 210, 286, 371
Insert a space after the numbering.
12) Line 257, 258
Correct the typo in the last column header.
13) Line 276
It is: „… for bearing:(a) Raw signals…”
Should be: „… for bearing: (a) Raw signals…”
14) Line 302
It is: „… three methods. (a) Dataset A, …”
Should be: „… three methods: (a) Dataset A, …”
15) Line 317
It is: „… results on Dataset A. (a) GAN, (b) …”
Should be: „… results on Dataset A: (a) GAN, (b) …”
16) Line 321
It is: „… results on Dataset B. (a) GAN, (b) …”
Should be: „… results on Dataset B: (a) GAN, (b) …”
17) Line 369
It is: „… results on Dataset C. (a) GAN, (b) …”
Should be: „… results on Dataset C: (a) GAN, (b) …”
Author Response
Response to Reviewer 2 Comments
Thank you for offering us an opportunity to improve the quality of our submitted manuscript. We appreciated very much your constructive and insightful comments. In the following text, our point-to-point responses to the queries raised are listed.
We highlighted all the revisions and responses in red color.
Thank you very much for the valuable suggestion. We have made corrections in the text.Please refer to the attachment for specific modifications.
We tried our best to improve the manuscript and we appreciate for your warm work earnestly, and hope that the correction will meet with approval.
Thanks again for your comments and suggestions.

Reviewer 3 Report
The proposed paper titled Dynamic Simulation Model Driven Fault Diagnosis Method for Bearing under Missing Fault Type Samples describing Fault diagnosis; missing samples; dynamic simulation; generative adversarial networks; gradient normalization with the topics: Theoretical Background, System Framework and Model Training, Experimental Verification and concluding remarks based, a new method based on dynamic simulation model and WGAN-GN was proposed as a solution for fault diagnosis of missing experimental fault samples of bearing, according to the authors. Additionally, It was demonstrated that the proposed method could generate high-quality replacement samples with missing fault samples to some extent and helps to improve the accuracy of diagnosis, which showed that the simulated samples generated by WGAN-GN were effective and trust- worthy. The paper was well written with no mistakes, and it is easy to read
The paper is original and free of mistakes
In my opinion , the authors must include an analysis of the influence of noisy on the proposed approach .
The concluding remarks and future works need to be improved : The concluding remarks were supported by the data, and it was also consistent with the evidence ( but they need more details)
Author Response
Response to Reviewer 3 Comments
Thank you for offering us an opportunity to improve the quality of our submitted manuscript. We appreciated very much your constructive and insightful comments.
We highlighted all the revisions and responses in red color.
Point 1:In my opinion , the authors must include an analysis of the influence of noisy on the proposed approach .
Response 1: At the beginning, we also thought about the analysis about the influence of noisy. In this article the data generated by WGAN-GN compared to the original data, it is clear that the feature trends are very similar and even remove some noisy components that are bad for pattern recognition. However, the problem to be solved in this paper is fault diagnosis for bearings when several types of fault signal data are missing. As for the problem about the influence of noisy, that is our part of future work to study. That is how to use neural networks more efficiently for noise reduction and feature extraction of vibration signals.
Point 2:The concluding remarks and future works need to be improved : The concluding remarks were supported by the data, and it was also consistent with the evidence ( but they need more details).
Response 2:Thank you very much for the careful suggestion, the revised conclusion section is shown as follow.
5.Conclusion
In this study, a new method based on dynamic simulation model and WGAN-GN is proposed as a solution for fault diagnosis of missing experimental fault samples of bearing. In the proposed method, a dynamic simulation model is first built to obtain simulated samples of missing fault type, which can be used to form relatively complete fault samples with other signals. Then WGAN-GN is employed to generate expanded samples from a smaller number of relatively complete fault samples. Next, these simulated samples are combined with the original samples to form an enhanced fault dataset which can be called as complete fault dataset. Finally, SAE is used to classify the complete fault samples. Three datasets of rotating machines were used to confirm the validity of the method. In addition, the effective feature extraction and classification capability of the proposed model is illustrated by the features learned continuously from the hidden layer of WGAN-GN. It is demonstrated that the proposed method can generate high-quality replacement samples with missing fault samples to some extent and helps to improve the accuracy of diagnosis, which shows that the simulated samples generated by WGAN-GN are effective and trustworthy. The following is a summary of the main conclusions:
(1) It is demonstrated that the developed dynamic simulation model can generate high-quality replacement samples with missing fault samples to some extent.
(2) The effective feature extraction and data generation capability of the proposed model is illustrated by the features learned continuously from the hidden layer of WGAN-GN.
(3) The experimental results show that using the proposed method can help to improve the accuracy of diagnosis when the types of fault sample data are insufficient.
(4) Applicability to other mechanisms and the problem about influence of noisy are two parts of our future research objectives. (Line 390-408, Page 15-16)
We tried our best to improve the manuscript and we appreciate for your warm work earnestly, and hope that the correction will meet with approval. Please refer to the attachment for specific modifications.
Thanks again for your comments and suggestions.

Reviewer 4 Report
In this work, a new method based dynamic simulation model and Wasserstein generative adversarial network with gradient normalization (WGAN-GN) to solve the missing fault type samples are investigated. The dynamic simulation model of bearing faults is constructed to obtaining simulation signals to replace and complement the missing fault samples. In addition, the employed gradient normalization (GN) module owns few hyperparameters and is easy to be applied to different rotating machines without substantial modifications. To testify the effectiveness of the simulated samples, a fault classification model constructed by stacked autoencoders (SAE) is used to classify the enhanced dataset.
Some results in this work are novel. Moreover, they are incremental improvements of earlier results. The level of difficulty and originality of these results makes them suitable for publication. The experts in the field of Dynamic Simulation Models will appreciate some technical progress exposed in this work. But I have some minor comments
· The abstract must be rewritten to show clearly the new results only and the first five lines can be written in the introduction section, if it is necessary.
· However, the introduction is good, but I think that it can be enhanced with adding and explaining some of recent related work
· The resolution of Diagrams must be improved, (In particular, Figs. 1,2, 7, 9, 12)
· The labels of axes must be included
· The authors have to shed light on the similarities and differences among their work and the literatures of the problem. A clear explanation, what is the new result in their work, and how it is build up upon previous work in the field.
· The translation of figures should be included properly
· A conclusion section should be extended to include more details
· In the last Section of the paper, I suggest adding a list of the main findings of the numerical analysis
In general, the authors have to describe in more detail the purpose of their study and its original contents.
If the authors submit a modified version according to my suggestions where they also give more details/explanations about the abovementioned criticisms, I could recommend the paper for publication.
Author Response
Response to Reviewer 4 Comments
Thank you for offering us an opportunity to improve the quality of our submitted manuscript. We appreciated very much your constructive and insightful comments. In the following text, our point-to-point responses to the queries raised are listed.
We highlighted all the revisions and responses in red color.
Point 1:The abstract must be rewritten to show clearly the new results only and the first five lines can be written in the introduction section, if it is necessary.
Response 1: Thank you very much for your valuable and professional suggestion, the revisions about abstract have been supplemented and marked with red as follows.
Abstract: Generative adversarial networks (GANs) can be used to effectively increase the volume of unbalanced samples. Although the existing generative adversarial networks (GAN) have the potential for data augmentation and intelligent fault diagnosis of bearings. However, the training instability and convergence difficulty of traditional GANs have detrimental effect on fault diagnosis. Moreover, most relevant studies only focus on fault diagnosis of rotating machines with sufficient fault type samples, but some rare fault type samples are may be missing for training in practical engineering. To address those deficiencies, this paper presents an intelligent fault diagnosis method based dynamic simulation model and Wasserstein generative adversarial network with gradient normalization (WGAN-GN) to solve the missing fault type samples. The dynamic simulation model of bearing faults is constructed to obtaining simulation signals to replace and complement the missing fault samples, which are combined with the measured signals as training data
and then input into the proposed WGAN-GN model for expanding and enhancing the data. In addition, the employed gradient normalization (GN) module owns few hyperparameters and is easy to be applied to different rotating machines without substantial modifications. To testify the effectiveness of the simulated samples, a fault classification model constructed by stacked autoencoders (SAE) is used to classify the enhanced dataset. According to the results, the proposed model performs well when used to fault diagnosis under missing samples and is preferable to other methods.
Point 2:However, the introduction is good, but I think that it can be enhanced with adding and explaining some of recent related work.
Response 2: Thank you very much for the careful suggestion, the revisions about the introduction have been supplemented and marked with red as follows.
Goodfellow et al. [11] proposed the generative adversarial network (GAN) that can generate new samples in an unsupervised learning way to extract the distribution properties of data. After that, GAN is also widely used in the field of fault diagnosis. Wang et al. [12] used GAN to synthesize fault signals from the spectra to expand the training volume and improved the fault classification accuracy of gearboxes. Han et al. [13] [12] introduced the adversarial learning mechanism into convolutional neural networks (CNN) and improved the fault classification accuracy of gearboxes. Li et al. [14] developed a modified auxiliary classifier GAN (MACGAN) that can generate multi-mode fault samples which are of superior quality. Han et al. [12] develops a novel framework for imbalanced fault classification based on Wasserstein generative adversarial networks with gradient penalty (WGAN-GP). Li et al. [13] developed a new ACGAN framework by adding an independent classifier. They introduced the Wasserstein distance and spectral normalization (SN) in the loss functions of GAN. Shao et al. [14] introduced the attention module to guide WGAN-GP to enhance the learning ability. It can be seen that GAN-like models have significant advantages in fault identification and diagnosis due to their characteristics.
It is observed that all of the above are aimed at generating more trustworthy sample data and enhancing training datasets by optimizing and improving the structure of the GAN when all the fault type samples are available for training. It is observed that most fault diagnosis methods can achieve better diagnostic results when all the fault type samples are available for training. However, sometimes the missing fault type samples may happen sometimes several types of fault signal data may miss, leading to failure to identify rare fault categories in practical engineering applications. Hence it is essential to build a model to replace and complement the missing fault samples.(Line 41-56, Page 1-2)
Point 3:The resolution of Diagrams must be improved(In particular, Figs. 1,2, 7, 9, 12).
Response 3: Thank you very much for the valuable suggestion, the figure Images have been recreated and increased in resolution. Please refer to the attachment for specific modifications.
Point 4:The labels of axes must be included.
Response 4: The labels of axes have be included.
Point 5:The authors have to shed light on the similarities and differences among their work and the literatures of the problem. A clear explanation, what is the new result in their work, and how it is build up upon previous work in the field.
Response 5: About previous work in this field, Han et al. [1] develops a novel framework for imbalanced fault classification based on Wasserstein generative adversarial networks with gradient penalty (WGAN-GP). Li et al. [2] developed a new ACGAN framework by adding an independent classifier. They introduced the Wasserstein distance and spectral normalization (SN) in the loss functions of GAN. Shao et al. [3] improved loss function is designed based on Wasserstein distance and gradient penalty to stabilize the training. The attention module is introduced to guide GAN to enhance the learning ability. All of the above are aimed at generating more trustworthy sample data and enhancing training datasets when there is only a tiny amount of sample data.
However, in our paper we wanted to address fault diagnosis for bearings when several types of fault signal data are missing instead of only simply generating and enhancing data. We established the dynamic model of the rolling bearing and obtained the missing fault types simulation vibration signals of rolling bearing. In addition, we firstly introduced the gradient normalization (GN) and Wasserstein distance into the GAN for fault diagnosis, which can also generate sample data with higher qualities. We believe that can provide some new methods and ideas for later researchers.
In other words, both the problem to be solved and the method proposed in this paper are different from the previous one. But the previous articles still provide us with valuable experience and reference, including the steps of their study and the experimental methods used, etc.
[1]Han, B.; Jia, S.; Liu; G.; Wang, J. Imbalanced fault classification of bearing via wasserstein generative adversarial networks with gradient penalty. Shock Vib. 2020: 2020,1-14.
[2]Li, W.; Zhong, X.; Shao, H.; Cai, B., & Yang, X. Multi-mode data augmentation and fault diagnosis of rotating machinery using modified ACGAN designed with new framework. Adv Eng Inform. 2022: 52, 101552.
[3]Shao, H.;Li, W.; Cai, B.; Wan, J.; Xiao, Y.; & Yan, S. Dual-Threshold Attention-Guided Gan and Limited Infrared Thermal Images for Rotating Machinery Fault Diagnosis Under Speed Fluctuation. IEEE T Ind Inform. 2023.
Point 6:A conclusion section should be extended to include more details.
Response 6:
5. Conclusion
In this study, a new method based on dynamic simulation model and WGAN-GN is proposed as a solution for fault diagnosis of missing experimental fault samples of bearing. In the proposed method, a dynamic simulation model is first built to obtain simulated samples of missing fault type, which can be used to form relatively complete fault samples with other signals. Then WGAN-GN is employed to generate expanded samples from a smaller number of relatively complete fault samples. Next, these simulated samples are combined with the original samples to form an enhanced fault dataset which can be called as complete fault dataset. Finally, SAE is used to classify the complete fault samples. Three datasets of rotating machines were used to confirm the validity of the method. In addition, the effective feature extraction and classification capability of the proposed model is illustrated by the features learned continuously from the hidden layer of WGAN-GN. It is demonstrated that the proposed method can generate high-quality replacement samples with missing fault samples to some extent and helps to improve the accuracy of diagnosis, which shows that the simulated samples generated by WGAN-GN are effective and trustworthy. The following is a summary of the main conclusions:
(1) It is demonstrated that the developed dynamic simulation model can generate high-quality replacement samples with missing fault samples to some extent.
(2) The effective feature extraction and data generation capability of the proposed model is illustrated by the features learned continuously from the hidden layer of WGAN-GN.
(3) The experimental results show that using the proposed method can help to improve the accuracy of diagnosis when the types of fault sample data are insufficient.
(4) Applicability to other mechanisms and the problem about influence of noisy are two parts of our future research objectives. (Line 390-408, Page 15-16)
We tried our best to improve the manuscript and we appreciate for your warm work earnestly, and hope that the correction will meet with approval.Please refer to the attachment for specific modifications.
Thanks again for your comments and suggestions.

Reviewer 5 Report
The paper presents the results of research related to the use of adversarial networks (GAN) for the diagnosis of machine faults. The authors proposed a rather interesting method of network training, but it is worth mentioning that a similar approach has been discussed many times in various publications. Nevertheless, the authors' considerations indicate the possibility of improving the quality of the created measurement systems. However, it is necessary to refer to the works of other authors in more detail, highlighting the existing problems.
Other work notes:
1. The authors at the beginning indicate that the presented method can be generalized, which allows for its wide application. However, the results presented in the paper concern only the ball bearing. Can the effectiveness of the method be confirmed for other mechanisms?
2. Have the obtained results been validated based on specific measurement data for other operating parameters of the bearings?
3. The conclusions are rather in the form of a short summary. I propose to indicate 3 - 4 specific conclusions that clearly follow from the conducted research.
Author Response
Response to Reviewer 5 Comments
Thank you for offering us an opportunity to improve the quality of our submitted manuscript. We appreciated very much your constructive and insightful comments. In the following text, our point-to-point responses to the queries raised are listed.
We highlighted all the revisions and responses in red color.
Point 1: The authors at the beginning indicate that the presented method can be generalized, which allows for its wide application. However, the results presented in the paper concern only the ball bearing. Can the effectiveness of the method be confirmed for other mechanisms?
Response 1: The introduced gradient normalization (GN) module did own few hyperparameters and is easy to be applied to different rotating machines without substantial modifications. However, in this paper we wanted to address fault diagnosis for bearings when several types of fault signal data are missing. Therefore, we only established the dynamic model of the rolling bearing and obtained the missing fault types simulation vibration signals of rolling bearing. Whether the GN method can be used for other mechanisms is also part of our future research objectives. And it has now been demonstrated in our recent work that it can be used for gears fault diagnosis .
Point 2: Have the obtained results been validated based on specific measurement data for other operating parameters of the bearings?
Response 2: In this paper, a dynamic model is established for simulation to obtain the missing bearing fault vibration signals, which are combined with the measured signals as training data and then input into the proposed WGAN-GN model for datasets enhancement, and the unprocessed measured true signals are used as testing data. To address fault diagnosis for bearings when missing several types of fault signal data. And the dynamic model only selects the SKF 6203 bearing as the simulation target bearing. The damage diameter is set to 0.3556 mm, the damage depth a is set to 2.794 mm, the rotational speed is 1797r/min, and the sampling frequency is set to 10 kHz.
Point 3: The conclusions are rather in the form of a short summary. I propose to indicate 3 - 4 specific conclusions that clearly follow from the conducted research.
Response 3: Thank you very much for the careful suggestion, the revised conclusion section is shown as follow.
5. Conlusion
In this study, a new method based on dynamic simulation model and WGAN-GN is proposed as a solution for fault diagnosis of missing experimental fault samples of bearing. In the proposed method, a dynamic simulation model is first built to obtain simulated samples of missing fault type, which can be used to form relatively complete fault samples with other signals. Then WGAN-GN is employed to generate expanded samples from a smaller number of relatively complete fault samples. Next, these simulated samples are combined with the original samples to form an enhanced fault dataset which can be called as complete fault dataset. Finally, SAE is used to classify the complete fault samples. Three datasets of rotating machines were used to confirm the validity of the method. In addition, the effective feature extraction and classification capability of the proposed model is illustrated by the features learned continuously from the hidden layer of WGAN-GN. It is demonstrated that the proposed method can generate high-quality replacement samples with missing fault samples to some extent and helps to improve the accuracy of diagnosis, which shows that the simulated samples generated by WGAN-GN are effective and trustworthy. The following is a summary of the main conclusions:
(1) It is demonstrated that the developed dynamic simulation model can generate high-quality replacement samples with missing fault samples to some extent.
(2) The effective feature extraction and data generation capability of the proposed model is illustrated by the features learned continuously from the hidden layer of WGAN-GN.
(3) The experimental results show that using the proposed method can help to improve the accuracy of diagnosis when the types of fault sample data are insufficient.
(4) Applicability to other mechanisms and the problem about influence of noisy are two parts of our future research objectives. (Line 390-408, Page 15-16)
We tried our best to improve the manuscript and we appreciate for your warm work earnestly, and hope that the correction will meet with approval. Please refer to the attachment for specific modifications
Thanks again for your comments and suggestions.

Round 2
Reviewer 1 Report
The authors considered the comments.